# High-Accuracy Event Classification of Distributed Optical Fiber Vibration Sensing Based on Time–Space Analysis

**DOI:** 10.3390/s22052053

**Published:** 2022-03-07

**Authors:** Zhao Ge, Hao Wu, Can Zhao, Ming Tang

**Affiliations:** Wuhan National Laboratory for Optoelectronics, School of Optical and Electronic Information, Huazhong University of Science and Technology, Wuhan 430074, China; m201977113@hust.edu.cn (Z.G.); zhao_can@hust.edu.cn (C.Z.); tangming@hust.edu.cn (M.T.)

**Keywords:** distributed vibration sensing, event classification, convolutional neural network

## Abstract

Distributed optical fiber vibration sensing (DVS) can measure vibration information along with an optical fiber. Accurate classification of vibration events is a key issue in practical applications of DVS. In this paper, we propose a convolutional neural network (CNN) to analyze DVS data and achieve high-accuracy event recognition fully. We conducted experiments outdoors and collected more than 10,000 sets of vibration data. Through training, the CNN acquired the features of the raw DVS data and achieved the accurate classification of multiple vibration events. The recognition accuracy reached 99.9% based on the time–space data, a higher than used time-domain, frequency–domain, and time–frequency domain data. Moreover, considering that the performance of the DVS and the testing environment would change over time, we experimented again after one week to verify the method’s generalization performance. The classification accuracy using the previously trained CNN is 99.2%, which is of great value in practical applications.

## 1. Introduction

The distributed optical fiber vibration sensing (DVS) technique can continuously measure a vibration signal along with an optical fiber [1,2,3,4]. It has the advantages of simple structure, small volume, corrosion resistance, electromagnetic interference resistance, and high sensitivity. In recent years, it has been applied in many fields, such as perimeter security [5,6], railway safety monitoring [7], pipeline monitoring [8,9], geophysical exploration [10], and so on. The classification of the vibration signals due to different events can increase the value of the DVS in practical applications. However, complex environmental noise and signal nonlinearity make it difficult for accurate event classification.

The DVS multi-class vibration event classification is usually divided into two steps: feature extraction and classification. Researchers first extracted information that can represent the characteristics of the data based on prior knowledge to reduce the difficulty of classification. The feature extraction was roughly performed through the methods of Mel-scale frequency cepstral coefficients (MFCC) [11,12], empirical mode decomposition (EMD) [13,14], fast Fourier transform (FFT) [15], and wavelet transform [16,17]. Unfortunately, these methods require manual optimization, which may not represent the vibration signals perfectly.

Recently, with the development of deep learning, convolutional neural networks (CNN) have been employed for DVS event classification due to their excellent feature extraction and classification ability. Xu et al. used the time–frequency diagram of the vibration signal as input data and used a two-dimensional (2D) CNN to extract the features in the time–frequency diagram for recognition [18]. However, this method cannot achieve end-to-end interference vibration recognition, and it has the problem of conflicting temporal and frequency resolutions. In addition, DVS signals are non-stationary signals, causing the interference signals to vary significantly from time to time, while the time–frequency domain data are more suitable for analyzing quasi-stationary signals. Wu et al. proposed a one-dimensional (1D) CNN classification scheme to analyze the time-domain data at the vibration point [19]. The dimensionality of the vibration feature analyzed by this method is single. The time-domain data type can only analyze data from the time dimension, but numbers of disturbing vibrations can cause a large range of spatial fluctuations. In 2020, the time–space data were used as input data, and the recognition rate was 97.1% [20]. However, its samples were discretely processed. The features of the time–space domain data and the ability of the CNN have not been fully explored. Therefore, there is still room for improvement.

In this paper, we propose a deep CNN-based method for DVS event classification. This method can achieve high-accuracy event classification for raw time–space data without redundant preprocessing. To demonstrate its advantages, we compared it with varied input data and neural network structures. We collected more than 10,000 sets of actual optical fiber vibration signals caused by hammer, air pick, and excavator. The experimental results, based on 1049 test samples, showed that the proposed method achieved an event classification accuracy of 99.9%. To further illustrate the method’s generalization performance, we collected 875 sets of data one week after the first experiment for a test. The accuracy using the previously trained CNN remained at 99.2%.

## 2. Data Set

### 2.1. DVS

We used the direct-detection phase–sensitive optical time-domain reflectometry (φ-OTDR) to achieve DVS because of its simple structure and low cost. The structure of the direct-detection φ-OTDR is like the optical time-domain reflectometry (OTDR) [21]. Pulsed light is injected into the sensing fiber, and then the backward Rayleigh scattered light generated during the propagation of the pulsed light is received. The difference is that the light source of the φ-OTDR is a highly coherent light source, which enhances the interference between the backward Rayleigh scattering. As shown in Figure 1, highly coherent light sources output continuous light. The semiconductor optical amplifier (SOA) modulates the continuous light into pulsed light. The pulsed light is amplified by an erbium-doped fiber amplifier (EDFA) and is injected into the sensing fiber through an optical circulator. The Rayleigh backscattering light is generated as the pulse propagates along with the fiber under test (FUT). The interferometric Rayleigh scattering signal is detected by an avalanche photodetector (APD) and then acquired by a data acquisition card (DAQ). The DVS system has a sampling rate of 80 million-sample/second (MSa/s), a spatial resolution of 5 meters (m), and a pulse repetition rate of 1 kilohertz (kHz). Therefore, the minimum time interval of the signal acquired by the system is 1 millisecond (ms), and the minimum spatial interval is 1.25 m.

### 2.2. Data Collection

The type of detection of the vibration event had several application scenarios, such as construction damage prevention in cities, perimeter security, earthquake disaster warning, etc. Different application scenarios are needed to identify different data types. The application scenario presented in this paper is the prevention of construction damage in cities, preventing underground cables from being accidentally dug by medium and large machines. The data presented in this paper are vibration event signals collected from outdoor experiments, including hammer strikes, air pick operation, and excavator operation. The air pick and excavator vibration events were the two typical events that most frequently occurred and most needed to be identified in the application scenario. The optical cable was 30 kilometers (km) long; near 20 km, a 50-m-long section of optic cable was buried 0.5 m below the soil. As the temperature and quality of the soil influenced the vibration signal [22], we randomly mixed stones and sand in the soil to increase the complexity of the environment. Moreover, we collected data from different days to study the impact of the equipment and environment changes over time. As shown in Figure 2, a hammer, an air pick, and an excavator were used to generate vibration above the optical cable. The hammer strikes had a short duration of action, while the air pick and excavator produced continuous signals. In addition, the vibration signal of the air pick was more regular compared to that of the excavator. However, due to the environmental noise and signal nonlinearity, the signals of different events constantly changed and were hard to be distinguished.

The data were first cut and normalized to increase the computational efficiency. Considering the number of the collected data and the characteristics of the three vibration events, we sliced the collected vibration data in a duration of 256 ms. Min–max scaling was chosen for normalization. The min–max scaling converted the original data linearization to the range of 0 to 1 and the normalization:$$X_{norm}=\frac{X-X_{\min}}{X_{\max}-X_{\min}}$$
where $X_{\max}\;$is the maximum and $X_{\min}$ is the minimum in the sample.

The space–time domain signals caused by the different disturbance events were preprocessed to form Dataset1, as shown in Figure 3. Each sample size in Dataset1 is 11 × 256, which contains 256 ms within a 12.5 m fiber length. The choice of sample size was based on the statistical analysis. More than 99% of the vibration durations and spatial fluctuation range were within the sample range. Figure 4 shows the statistical analysis of the spatial fluctuation range caused by vibration.

In order to study the impact of different data types on pattern recognition, we performed different processing on the same data to generate multiple datasets. We collected the 1D time-domain data at vibration points to form Dataset2. Shown in Figure 5a–c are the 1D time-domain data of hammer strikes, air pick operation, and excavator operation, respectively. Each sample of Dataset2 has a duration of 256 ms and contains 256 data points. Then, we performed the fast Fourier transform (FFT) on the time-domain data and normalized them to form Dataset3, as shown in Figure 5d–f. Moreover, the short-time Fourier transform (STFT) was applied to the time-domain data to generate the time–frequency data to form Dataset4, as shown in Figure 5g–i.

It is worth noting that Figure 3 and Figure 5 show only the typical feature of the interference event, and not all samples are like Figure 3 and Figure 5. More time-domain and frequency-domain data of the different vibration events are shown in Figure 6 and Figure 7, where Figure 6a–c and Figure 7a–c are hammer events, Figure 6d–f and Figure 7d–f are wind pick events, and Figure 6g–i and Figure 7g–i are excavator events. There are mainly three reasons for the differences in the features between the same events. Firstly, the φ-OTDR signals are non-stationary, so the signals change continuously with time. The samples of vibration events collected in this work were not collected at the same time or the same place. Therefore, there are certain feature differences between the signals collected at different times or locations. Secondly, the noise of the outdoor environment was also complex and variable, which also influenced the collected data. Moreover, the time of the sample slicing was fixed during the preprocessing process, so some vibration events may have occurred at the beginning or ending in a sample, causing the loss of some vibration features. All these factors can lead to a large difference between the signals of the same event.

We conducted two experiments one week apart as the environmental noise, the condition of the fiber, and the performance of the equipment changed over time. In the first experiment, 10,249 samples were collected and randomly divided into training sets: a validation set and Testset1, with 8250, 950, and 1049 samples, respectively. In the second experiment, we collected 875 samples to form the Testset2. The training set was used for neural network training. The validation set was used to determine whether the neural network was over-fitted during training, and the test sets were used for testing. The data of Testset1 were collected on the same day as the training set and the validation set, while the data of Testset2 were collected one week later at different locations. The first experiment was conducted to compare the classification accuracy using the four data types. The second experiment was conducted to investigate the generalization performance.

## 3. Deep CN

In this experiment, we used a deep CNN to identify and classify the vibration intrusion events. A typical CNN consists of a convolution layer (Conv), pooling layer (Pool), and fully connected layer (FC). Since a CNN needs to recognize the time–space domain signals without manual feature extraction, the original time–space joint features were more complex than the other signal features. There is a high requirement for its feature extraction capability. Generally, a deeper network can improve classification performance, including generalization performance and recognition accuracy [23]. However, the more network layers, the more difficult it is to train. Therefore, we introduced the ResBlock, whose structure is shown in Figure 8b [24]. It established a direct connection between the input and output, resulting in identity mapping so that the model focused on learning the residual between the input and output, thus speeding up the training and improving the network performance [24]. The CNN model, composed by stacking multiple ResBlocks, effectively improved the problem of the training difficulties caused by too many layers.

Figure 8a shows the structure of a 2D deep CNN for 2D time–space domain signals. In this work, the ReLU was used as the activation function, and the Adam was used as the optimization function. Additionally, the loss function was the cross-entropy loss function. The pooling strategies were the maximum pooling and adaptive mean pooling. The best model was chosen from the validation set with the lowest loss function, and training was stopped once the best model had run for 20 iterations without updates. In the early stage of the network, the convolutional and pooling layers were mainly used to perform basic operations, such as boosting and pooling the input data to accelerate the network’s training. Then, multiple ResBlocks extracted features from the data continuously. As the network deepened, the features learned became more abstract, so the number of channels for the ResBlocks needed to be increased to give the network more expressiveness and to cover as many key features as possible. Finally, the fully connected network mapped the features into a space where they could be further separated and output the probabilities of the different events through the softmax layer.

Four CNN models were constructed to establish the optimal network by changing the number of ResBlocks. The average recognition accuracies of Testset1 were verified, as shown in Figure 9. The CNN model with six ResBlocks appeared to be the best option. The architecture of the CNN model is displayed in Figure 10. For comparison, a 1D deep CNN with the same structure was built to process the 1D data, and the sizes of the 1D CNN convolutional kernel are indicated in brackets in Figure 10. Each convolutional layer contained a batch normalization (BN) layer and a nonlinear activation layer.

## 4. Results and Discussion

### 4.1. Training Process

The training sets of Datasets2 and Datasets3 were input to the 1D CNN, and the training sets of Datasets1 and Datasets4 were input to the 2D CNN, respectively. To compare the performance of the different methods, the learning rate, loss function, activation function, training batch, backpropagation algorithm, and the number of iterations of these four methods were all the same. The learning rate was 10^−3^, the loss function was cross-entropy, and the batch size was 64. The validation set was used to ensure that the neural network did not overfit during training. When the network was overfitted, continuing the iteration would reduce the model’s generalization ability, so we chose the model with the lowest validation loss among 100 epochs. Figure 11 shows the accuracy of the validation set with the number of iterations during the training process. It is normal for the validation accuracy to drop occasionally with the training epoch. The network may occasionally optimize in the wrong direction as the gradient decreases. However, our optimization method could quickly detect the error and find a new direction for optimization. The time–space domain data had the best identification accuracy in the validation set, while the frequency-domain data had the poorest, and the time–frequency data was similar to the time-domain data. Compared with the time-domain signal, spatial scale information was added to the time–space domain signal, so the recognition was more accurate. Although the time–frequency domain signals increased the frequency-domain information compared with the time–domain, the data before STFT is the raw data without additional preprocessing. Therefore, the recognition accuracy of the two is not much different.

### 4.2. Test Results

After training, Testset1 was fed into the corresponding trained neural networks, and the results are shown in Table 1. For the three vibration events, the recognition rate using the time–space domain data with the 2D CNN reached 99.90%, 1.62% higher than the second-best result using the time-domain data with the 1D CNN. The former only misidentified the hammer event once. The difference between these two models only depended on whether the space dimension information was introduced. The temporal information could better identify the vibration events with regular disturbances of long duration, such as the air pick, and the space information had a positive impact on recognizing the transient disturbance events and long-duration disturbance events. The recognition rate of the frequency-domain data was only 95.52% lower than 98.28% for the time-domain data and 97.52% for the time–frequency data. The 2D CNN successfully learned the features from the raw time–space domain data of various events, and the deep CNNs feature extraction capability was sufficient to recognize the complex raw data without additional preprocessing.

The DVS signal was a non-stationary signal, causing the interference signal to vary significantly from time to time. Therefore, the generalization performance of the trained models was critical. We used the same system to collect samples a week later to verify the generalization performance of the models to different environmental conditions and equipment conditions. The samples are the Testset2 mentioned in 2.2. We used the trained models to analyze Testset2. The results are shown in Table 2. The recognition rate using the time–space domain data was 99.20%, while the time-domain data’s recognition rate dropped to 95.65%. That of the frequency-domain data and the time–frequency domain data dropped to 93.03% and 93.20%, respectively. This is because the signal and noise varied randomly from time to time. The Testset2 was collected several days apart from the other datasets, and some of them had changed in their abstract features that had not been learned in the training set. The time-domain data model could only learn the feature changes of a single vibration point from the time dimension, and the recognition was limited compared to the joint space–time features. Therefore, the data association between the adjacent spaces promoted the system’s greater generalization ability. In practical applications, the real-time nature of the system was also very important. After testing, the model proposed in this paper took 140 ms to process once, which met the need of real-time.

### 4.3. Comparison of Neural Networks

As mentioned in Section 3, the deeper CNN could provide higher accuracy and generalization ability compared to shallow CNNs. In order to better demonstrate the performance of the deep CNN model proposed in this paper, we compared it with the CNN proposed in the reference [20]. This CNN contained four convolutional layers and two pooling layers, while our CNN contained 13 convolutional layers and two pooling layers. The time–space data in Dataset1 were input into two different structured neural networks for training. Their accuracy was compared through Testset1, and the generalization ability was analyzed using Testset2. The results are shown in Table 3. The shallow CNN achieved a 98.76% recognition rate for the data in Testset1, but the recognition rate in Testset2 dropped to 65.83%. In contrast, our CNN had deeper convolutional layers, which made the recognition rate of Testset2 reach 99.20%. The powerful feature extraction ability of the deep CNN ensured that it could accurately learn the features of the corresponding events from the complex interference and enhance the model’s generalization ability.

## 5. Conclusions, Limitations, and Future Research Trends

In this paper, we proposed a DVS event classification method based on the original time–space domain signal and a 2D deep CNN. We used a 2D deep CNN to automatically extract features from the original vibration signal without manual feature extraction. The results of the real experimental data show that the time–space domain data model has a 99.9% recognition rate in Testset1, which is similar in composition to the training set. The recognition rates of the data models for the time-domain, frequency-domain, and time–frequency domain on Tesset1 were 98.3%, 95.5%, 97.5%, respectively. In Testset2, collected one week later, the time–space domain data type still had more than 99% recognition, while the recognition rate of the other models dropped to 95.7%, 93%, and 95.5%, respectively. Compared with CNNs with fewer layers, the deep CNNs could significantly enhance the learning ability of the model on the original time–space domain data, and the proposed deep CNN took 140 ms to process once, which met the requirement of real-time.

Although the proposed method had a stronger generalization ability than the other methods, its accuracy on the data collected one week later still dropped. To adapt to the constantly changing environment, we will study an automatic optimization method based on reinforcement learning to improve its performance further.

## Figures and Tables

**Figure 1 sensors-22-02053-f001:**
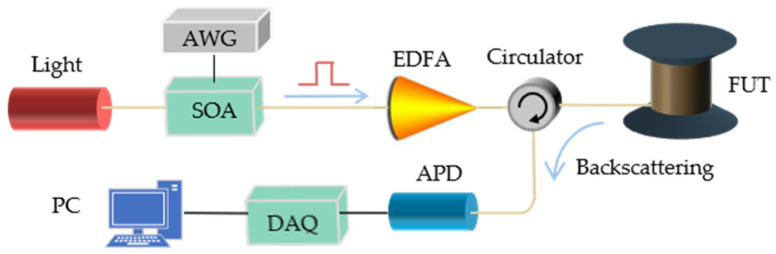
Schematic diagram of DVS system. SOA: semiconductor optical amplifier, AWG: arbitrary waveform generator, EDFA: erbium-doped fiber amplifier, FUT: fiber under test, APD: avalanche photodetector, and DAQ: data acquisition card.

**Figure 2 sensors-22-02053-f002:**
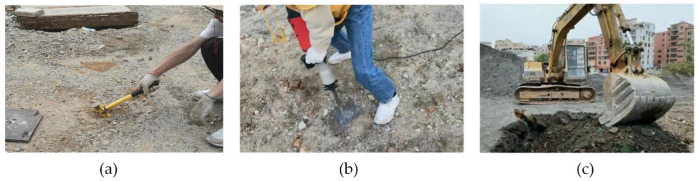
Field tests for collecting vibration signals: (**a**) hammer, (**b**) air pick, and (**c**) excavator.

**Figure 3 sensors-22-02053-f003:**
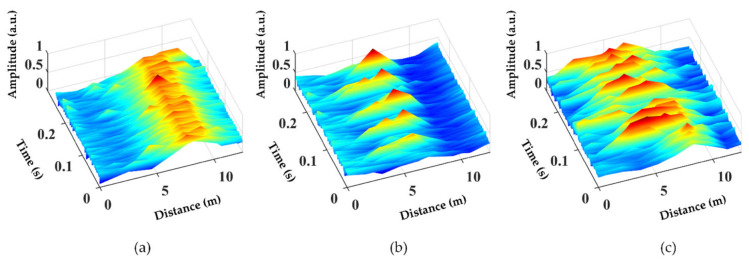
Time–space domain diagrams of three typical disturbance events: (**a**) hammer, (**b**) air pick, and (**c**) excavator.

**Figure 4 sensors-22-02053-f004:**
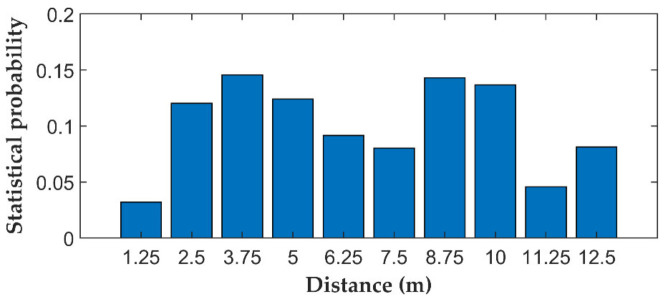
The statistical analysis of the spatial fluctuation range caused by vibration.

**Figure 5 sensors-22-02053-f005:**
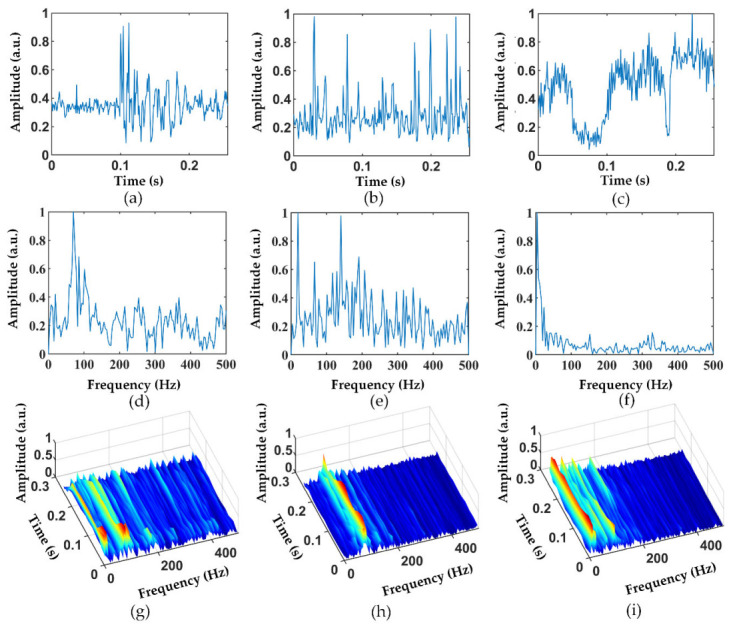
(**a**–**c**) Time-domain diagrams of a hammer, air pick, and excavator; (**d**–**f**) frequency-domain diagrams of a hammer, air pick, and excavator; and (**g**–**i**) time–frequency domain diagrams of a hammer, air pick, and excavator.

**Figure 6 sensors-22-02053-f006:**
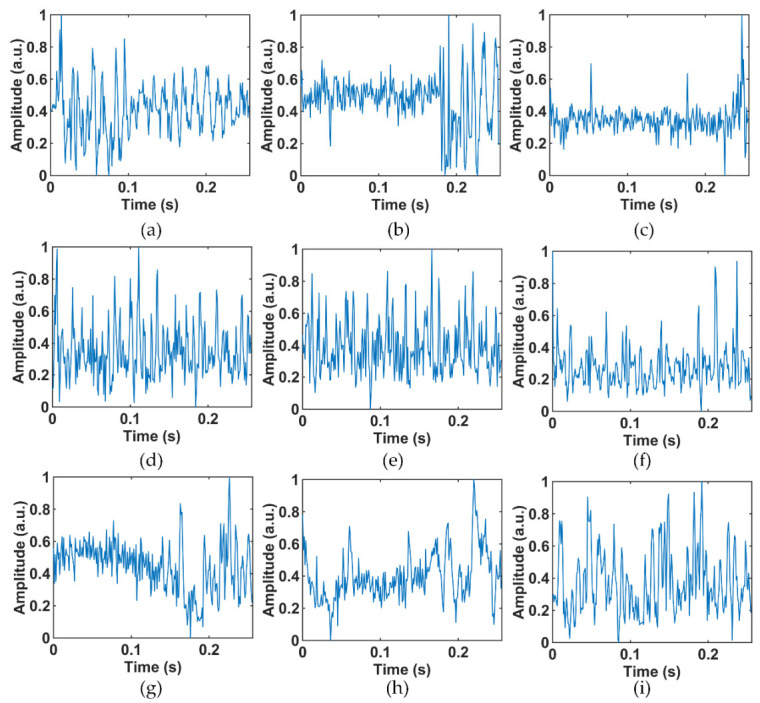
Other samples in the time-domain dataset: (**a**–**c**) hammer; (**d**–**f**) air pick; and (**g**–**i**) excavator.

**Figure 7 sensors-22-02053-f007:**
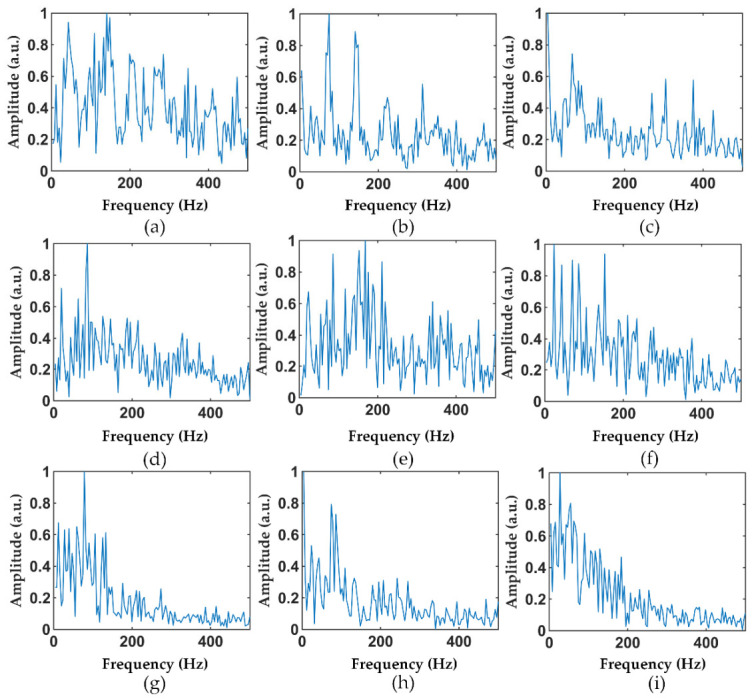
Other samples in the frequency-domain dataset: (**a**–**c**) hammer; (**d**–**f**) air pick; and (**g**–**i**) excavator.

**Figure 8 sensors-22-02053-f008:**
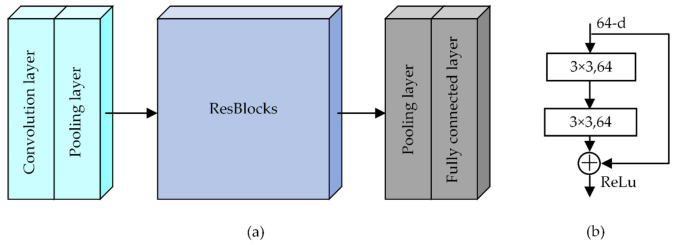
(**a**) Basic architecture of a 2D CNN; (**b**) the structure of the ResBlock.

**Figure 9 sensors-22-02053-f009:**
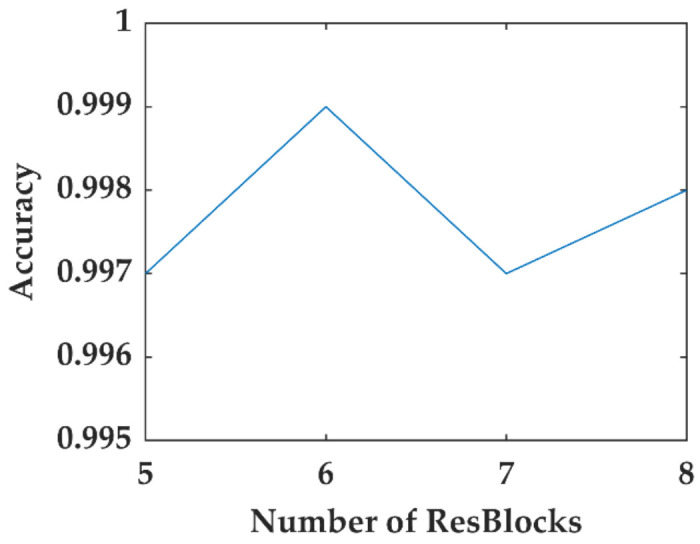
Accuracy of different numbers of ResBlocks in Testset1.

**Figure 10 sensors-22-02053-f010:**
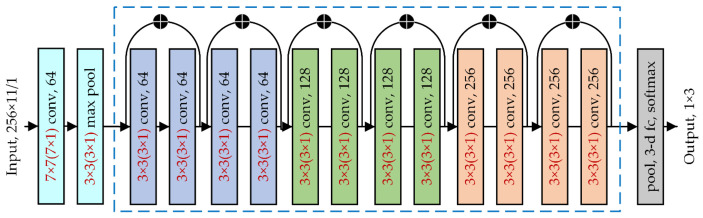
Structure of the deep 1D CNN and 2D CNN.

**Figure 11 sensors-22-02053-f011:**
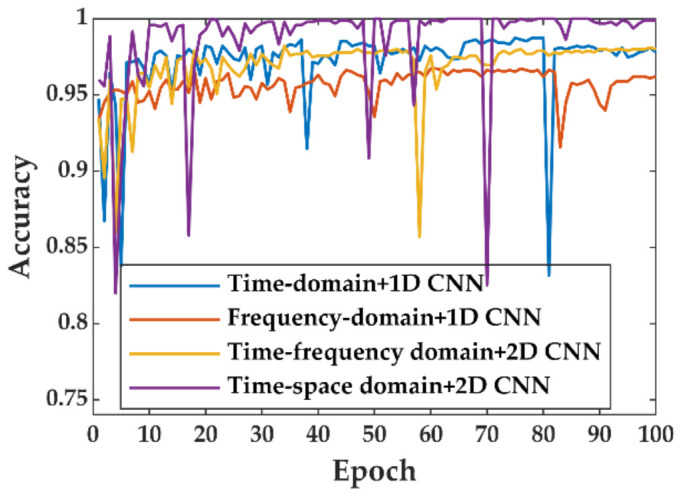
The accuracy curves of each epoch validation dataset.

**Table 1 sensors-22-02053-t001:** The event classification accuracy of Testset1.

Class	Identify Category/%			Recognition Rate/%
	Hammer	Air Pick	Excavator	
Time–space domain + 2D CNN	99.70	100	100	99.90
Time-domain + 1D CNN	97.59	99.44	97.77	98.28
Frequency-domain + 1D CNN	94.44	97.26	94.74	95.52
Time–frequency domain + 2D CNN	96.61	99.47	96.26	97.52

**Table 2 sensors-22-02053-t002:** The event classification accuracy of Testset2.

Class	Identify Category/%			Recognition Rate/%
	Hammer	Air Pick	Excavator	
Time–space domain + 2D CNN	97.76	99.70	100	99.20
Time–domain + 1D CNN	93.66	94.85	98.56	95.65
Frequency-domain + 1D CNN	97.76	85.45	97.47	93.03
Time–frequency domain + 2D CNN	97.76	93.03	96.39	95.54

**Table 3 sensors-22-02053-t003:** Comparation of neural networks.

Class	Recognition Rate of Testset1	Recognition Rate of Testset2
Our CNN	99.90%	99.20%
CNN [20]	98.76%	65.83%

## Data Availability

Not applicable.

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
