# Peer review of "High-Accuracy Event Classification of Distributed Optical Fiber Vibration Sensing Based on Time–Space Analysis"

_sensors, 2022, doi:10.3390/s22052053_

Round 1
Reviewer 1 Report
In the manuscript, the authors experimentally demonstrate a pattern recognition method with high accuracy for DVS. The results are impressive and meaningful for the practical application. The manuscript can be considered for publication after the following points are addressed:
- The experimental conditions need to be introduced more clearly, such as the length of the optical fiber buried in the ground, the location of the vibration, etc.
- In Figure 4, the time-domain signals of different events are clearly different. Why can't the neural network distinguish them accurately?
- Why are the number of channels different for different Resblocks? Is it fair to compare the effect of the number of Resblocksin this case?
- The reasons for choosing 256ms duration and 12.5m fiber length are not clear enough. Could a stronger justification be given by statistical analysis of the data?
- In practical applications, processing speed is a key parameter. Can the proposed algorithm be computed in real time?
- The electrical pulse signal generator is missing in Fig.1.
Reviewer 2 Report
The authors investigated an event classification of distributed optical fiber vibration method. Although the authors obtained excellent results, considering the scholarly contribution, publication is unfortunately not recommended.
1. φ-OTDR is a commonly used optical fiber sensing structure, and CNN is a commonly used neural network structure. The combination of the two fails to reflect innovation.
2. The type detection of vibration has certain application value. However, the introduction of this paper does not reflect the purpose of distinguishing the above three types of data, nor can it describe the insufficiency of research in related fields.
3. The above three signals can be clearly distinguished from Figure 3-4. If only the above three characteristics are distinguished, it can even be judged directly from the frequency domain. Therefore, obtaining excellent results is a must. For example, in Table 1, a simple temporal + 1DCNN achieves over 98% results. Although the author obtained more than 99% of the results, its significance is not significant.
4. Although the experiments in this paper provide a lot of data, they are limited to three types of mutual judgments. The authors should provide a wider variety of signals, such as walking, driving, stomping, impacting, etc., from which it may be more meaningful to be able to discern behaviors such as excavator.
Reviewer 3 Report
This paper has proposed a deep learning convolutional neural network (CNN) method to classify DVS (distributed optical fiber vibration) experimental data that had been done outdoors, and finally, the time-space CNN classification has achieved really high accuracy.
This topic would be attractive for the reader in DVS and Deep Learning community, and this paper is well written.
I recommend editor accept this manuscript with the following revisions.
1. In Line 42, "2-dimensional (2D)" could be "two-dimensional (2D)".
2. In Line 62, "OTDR" should be followed by full name, which is different from "phi-ODTR".
3. In Lines 105 - 120, "256-ms" could be "256 millisecond (ms)", "different data type" should be "different data types",
"which contains 256 data" could be "and contains ...".
4. In Line 74, the abbreviation 'MSa/s' should be specified, or followed by a full name when it appears for the first time. Please also correct other units or abbreviations throughout the paper.
5. In Lines 128 - 133, these sentences are a bit wordy, please make it concise.
6. In Figure 5.b, "relu" should be corrected as "ReLu". And please specify a bit "ResBlock" in Section 3 when it is first introduced.
7. In Figure 8, the label of the y-axis should be "accuracy". "STFT" should be defined.
And what causes the accuracy to drop as shown in Figure 8? Will they repeat after the number of epochs goes beyond 100?
8. In Section 4.3, please add quantitative comparison results in the discussion, instead of only "good" and "better".
By the way, why do you choose the current neural network structure of CNN? Is it determined by optimization or sensitivity analysis? Please specify that.
9. In Line 222, "as mentioned before", please indicate the detailed insection instead of only "before".
10. The conclusion looks like the abstract, please add more quantitative analysis results that support the final conclusions.
11. In conclusion, please specify the limitation of this work and the future perspectives.
Round 2
Reviewer 2 Report
After viewing the modified version, I recommend its publication.